# A Digital Twin Case Study on Automotive Production Line

**DOI:** 10.3390/s22186963

**Published:** 2022-09-14

**Authors:** Arif Furkan Mendi

**Affiliations:** 1HAVELSAN, Information and Communication Technologies, 06510 Ankara, Turkey; afmendi@havelsan.com.tr or ariffurkan.mendi@ostimteknik.edu.tr; 2Department of Computer Engineering, Ostim Technical University, 06370 Ankara, Turkey

**Keywords:** digital twin, manufacturing, IoT, IIoT

## Abstract

The manufacturing sector is one of the areas where the advantages of digital twin technology can benefit mostly. The product development, including its software, electronics, mechanics, and physical behavior, is included in the digital twin of the product. Furthermore, simultaneous data capturing from the sensors and data processing are also available in the digital twin. This enables each phase of the development cycle to be simulated, processed, and validated to discover the potential problems before the production of real components. In this study, the use of digital twin technology in the commercial production phase of the automotive production line with a case study is introduced. This study is one of the most comprehensive studies in the literature related to automotive production; therefore, it puts forth the power of using digital twin technology in that area. As the result of this case study, a 6.01% increase in the commercial production line efficiency and an 87.56% gain for downtime are achieved.

## 1. Introduction

Digitalization has begun to be a part of our daily life after the maturation and spread of emerging technologies. One of the main reasons for digitalization is to reduce or even eliminate the need for human power [1]. With the widespread use of the Industry 4.0 approach, it is predicted that 30% of today’s jobs, including white collar positions in all departments, will be completed by robots [2]. Specifically, scientific progress over the last few years shows that the social environment for production has changed significantly, and has increased global market competition and the diversity of customer demands. Responding to the changing environment, the manufacturing industry and related businesses pay more attention to certain manufacturing features [3]. We see that an innovative digital transformation journey has already started due to the developments in the Internet of Things (IoT) technology and the Industry 4.0 revolution, which have become more important with digitalization [4]. Digital twin (DT) technology, which is based on the digitization of physical systems, has an important place as a keystone technology in this journey. The DT technology defined as a virtual model of the behavior and results of a physical product or a service in the real world was introduced in 2002, but its popularity has been increasing in recent years with the widespread use of the IoT technology [5]. A DT is not just an electronic copy of a physical entity, such as an object, process, or person. Moreover, it also includes digital (or virtual representation) of the physical entity. DTs are dynamic, virtual models of the physical world, enhancing our ability to understand, learn from, and reason from changes in the environment. DTs are the foundation of smart applications that continuously collect sensor data to optimize performance, predict errors, and simulate future scenarios. If DT technology is adapted and solutions are developed with this technology, rapid increases in the development level of countries will be seen with the advantages it offers to producers and consumers in terms of cost, time, and quality.

The Fourth Industrial Revolution drives industries from traditional manufacturing to the smart manufacturing approach. In this transformation, existing equipment, processes, or devices are retrofitted with some sensors and other cyber-physical systems (CPS), and adapted toward digital production, which is a blend of critical enabling technologies [6]. CPS is a new trend in IoT-related research works, where physical systems act as sensors to collect real-world information and communicate them to the computation modules (like the cyber layer), which further analyze and notify the endings to the corresponding physical systems through a feedback loop. In the current scenario of Industry 4.0, industries are shaping themselves towards the development of customized and cost-effective processes to satisfy customer needs with the aid of a digital twin framework, which enables the user to monitor, simulate, control, optimize, and identify defects and trends within the ongoing process, and reduces the chances of human prone errors [7]. The core element of this trend is mostly related to DT. In order to gain the advantages of digital twin technology, it is not enough to simply adapt the technology. It is difficult and important to draw the architecture correctly [8].

DTs are created using simulation, sensors, IoT, big data, and artificial intelligence technologies. Sensor, simulation, and IoT technologies are in digital modeling of physical assets; big data, machine learning, and artificial intelligence are used to analyze the data collected for digital twins, learn the behavior of the entity, and make predictions for the future. Due to the development of these technologies used, digital twin technology is also positively affected by developments. While creating a DT, Computer-Aided Technologies (CAx) and sensor data are provided through Industrial Internet of Things (IIoT) technology, while cloud and edge computing technologies are used while processing or before processing these data, artificial intelligence techniques to make sense of the data, and finally, big data analysis and data visualization processes are being carried out. All these processes can be performed throughout the life cycle of the product. As a result, we can easily say that six different technologies can be used to create a digital twin (Figure 1).

It is seen that this technology, which is likely to be used in many areas of our lives, attracts great attention from technology developers, users, and investors. In an estimate of the benefits of digital twins, IDC has suggested that the use of this technology could increase the speed of critical processes by 30% [9]. Digital twin technology comes to the forefront by increasing the speed of critical processes and many other advantages. For this reason, many investors prefer this technology. According to MarketsAndMarkets’ “Digital Twin Market by Technology, Type, Industry and Geography-Global Forecast to 2026” report, while the digital twin technology market is 3.1 billion dollars in 2020, it is expected to reach $48.2 billion by 2026, with an annual growth rate of 58% [10]. This shows us concretely how the digital twin technology market will become important soon. When we examine the popular application areas of digital twin technology, we see that the manufacturing sector is the most popular area, followed by smart city applications, platform and other applications, energy, agriculture, and health applications. According to the “Industrial Distribution of the Worldwide Digital Twin Market in 2020” research conducted by Grand View Research, the production sector is the sector in which digital twin technology is used the most, with a rate of 39% (Figure 1).

The contributions of this proposed study are:It is the first case study in which efficiency increase and false alarm reduction are achieved with the use of DT technology in an extensive automotive production line.Besides efficiency, downtime (false alarms) is also proved to be reduced practically by using DT.With the very promising efficiency and downtime results of this study, it is shown that electricity saving is achieved with high gain, which is a good contribution made to the provision of a green society by preventing energy waste.

This paper introduces a case study conducted on DT in the manufacturing sector. When choosing the automotive sector, the aim is adding improvements on manufacturing lines to speed up the process and increase efficiency. The rest of the paper is organized as a literature review related to DT and its applications in the automotive sector, materials, method and system design, results and discussion, and finally the conclusion.

## 2. Literature Review

The product idea, product design, manufacture plan, product involvement, maintenance, and product renewal are all areas where DT plays an important role. With the extension to manufacturing, prognostics, design, and health management [12,13,14], it is commonly used to increase efficiency, improve information flow, and automate procedures [15]. DT models can be used for maintenance, enhancing the speed of product tracking, or virtualizing particular circumstances of the product throughout its activities during the operating phase [16].

DT is an important tool in the continuous improvement of the production system. The study in [17] shows that big manufacturing facilities and factories may generate DT models to optimize working area layouts and improve safety, efficiency, and ergonomics. Furthermore, smart manufacturing has demonstrated increased degrees of automation and adaptability to altering material combinations to maintain their processes [18,19]. The study in [20] suggested using a machine learning-based DT system to optimize the production process in petrochemistry. DT, in conjunction with IoT, machine learning (ML), and signal processing technologies, is critical to the shift from traditional production to smart manufacturing. This concept has the potential to benefit industrial industries by increasing flexibility and automation in numerous processes [19]. It is also seen as a vital option for improving digital surveillance systems and the operation of adaptive and more automated systems’ networked components [21,22]. Many firms have considered DT to be a vital tool for improving their processes. The integration of DT into production systems was discovered to be a solution to the increasing complexity of the order management process, as well as to increase the firms’ flexibility and profitability [23]. Furthermore, this technology is thought to be a novel way to offer continuous digital control and active functional improvement of networked items and equipment. This is a potential strategy for businesses to respond to changing client demands, greater uncertainty, and higher resource costs.

The DT model may be used to anticipate future asset behavior and repercussions due to disruptions using various modeling methodologies such as simulation-based, data-based, or mathematical modeling. As a live model, DT must recognize possible issues with its physical twin. DT can give reliable predictions for particular predictive maintenance utilizing continually obtained data and the industrial Internet of Things [23,24,25,26,27]. As a result, by providing a mirror of the actions of a physical twin, DT may play a significant role in the warning system, forecasting, and management of a manufacturing system or service. Because many businesses are shifting from reactive to proactive maintenance to decrease operational loss, maintenance costs, and capital investment, the DT model has become increasingly important in this sector. The significance of DT technology in aviation maintenance operations, as well as the influence of technological breakthroughs and needs on the traditional sector, have been described in the research [25,28].

One of the challenges with digital twins is collecting data from physical systems. Data collection for digital twins in cyber-physical systems, where the controllers are often hard and in real-time, is a difficult issue, as the functional system behavior may be impacted by the instrumentation. A frequently implemented solution is the use of additional monitoring controllers that passively monitor the system. However, this solution introduces additional points of failure and is costly as more hardware is required. In order to solve this problem, Mertens and others proposed a new approach. They proposed an approach that aids the digital twin developer by finding the optimal instrumentation rate for a set of parameters that are to be instrumented on a hard real-time controller. The optimization ensures that the hard real-time constraints are guaranteed, while also taking into account the developer’s preferences about the instrumented parameters. This process is applied to two cases: an exploratory example and an adaptive cruise controller. The exploratory example showed some of the drawbacks of the solver, and gave more insight into its operation, whereas the ACC case study showed that more difficult problems can also be solved [29].

As a result of the widespread learning of the opportunities provided by the digital twin technology by the societies and especially the producers, we see that the use of this technology has increased on a global scale and has begun to find a place for itself in many areas of manufacturing. Many companies currently use DT for their processes.

For instance, Unilever has applied its digital twins’ concept to scale-up consistency in soap and detergent production, reduced the number of false alerts that require action in its facilities, and announced a 90% reduction in false alerts [30].

With KINEXON’s position-based digital twin automation concept for automatic tool control along the automotive assembly line, it is declared that it created a 5% faster assembly line. It has also been announced that the number of manual errors and product recalls has decreased [31].

In the “Digital Twin Genie” case study conducted in China, a digital twin solution was implemented to minimize production and development costs in the automobile manufacturing industry by using the digital twin platform, which monitored machine temperature, performance speed, power input–output, etc. The data were collected and monitored by a large number of sensors, both analog and digital, and with the various capabilities of the machine learning engine used. As a result of the study, production and development costs were reduced by 54%, the total downtime of the machines was reduced by 37%, and the average production time was reduced from 14–17 h to 9–10 h [32].

In order to develop China’s building materials industry, a $135.96 million project has been initiated to establish a world-class, green, smart cement factory with a daily production capacity of 4500 tons. For this purpose, Citic Heavy Industries, using iTwin for the digital twin infrastructure of the factory and production lines, provided visual monitoring of the cement production process by associating the 3D equipment model with smart digital data processes. As a result of the study, more than 30% savings in operation and maintenance costs were achieved and a preventive asset maintenance system was implemented, which reduced equipment maintenance costs by $2 million compared to a planned asset management process. In addition, it was emphasized that smart, green factory production processes in the future are expected to reduce annual electricity consumption by 3.5 million kWh [33].

Horvathova claimed digital twin could be used to improve the efficiency of leather cutting in the automotive industry. The authors proposed alternatives of increasing the efficiency of material selection and processing in the selected company and reducing costs and leather sustainability as a result. As a result, by the use of digital twin and other Industry 4.0 principles and solutions in the process of material selection and processing in the company selected, increased efficiency and cost savings were achieved [34].

Similarly, Florian Biesinger and his colleagues emphasized the necessity for a digital twin of production systems in the automotive industry. A survey on the use of a digital twin of the production line was conducted with the production planning department of a major automobile manufacturer by them. That study identified exactly the information required by a digital twin of the production plant to improve the integration planning process. Moreover, this survey provided answers to the benefits of integration planning by an automatically generated digital twin [35].

Stavropoulos and the others also conducted a study on a molecular dynamics-based digital twin for ultrafast laser material removal processes. The process of material removal utilizing Femto lasers has been examined both theoretically, with the use of molecular dynamics-based simulations, and experimentally. The experimental data from the Femto laser ablation have been compared with the simulation results, and the applicability of the digital twin model has been evaluated [36].

Yoo Ho Son and his colleagues proposed work on a digital twin-based cyber-physical system for automotive body production lines. Unlike previous research on digital twins focusing on independent engineering application development, they designed and implemented a CPS combined with a digital twin and other components for a web-based integrated manufacturing platform. They claimed that this is the first time a digital twin-based CPS was implemented for abnormal scenarios involving automotive body production lines; the capability of the proposed system was verified via experiments. The experimental results indicated that the proposed system achieved an average prediction performance of 96.83% for the actual production plan. They added that the digital twin-based CPS can be applied to automotive body production lines, and it provides an advanced solution to predict whether production is possible according to the production plan [37].

When these exemplary studies, especially in the production and automotive fields, are evaluated, it is seen that digital twin technology can be an element that will increase efficiency in the automotive production line.

## 3. Materials, Methods, and Applications

In this section, the materials used within the scope of the study, and the method by which these materials were used and the study was carried out will be explained.

### 3.1. Materials

Different sensors and measuring devices were used within the scope of the study. The E52-ELP6-50-2-A temperature sensor was used for receiving temperature data from the physical system, obtaining status from the Kuka KSS interface, and the temperature of robotic arms. A Telaire Smart Dust Sensor SM-PWM-1C was used for measuring the dust density. An Arduino-based tachometer frequency sensor was used for obtaining the actual CNC frequency value to hold the frequency in the optimal range (Table 1).

Various software has been used for the transmission, storage, processing, and visualization of the data obtained through the sensors. The obtained sensor data were transmitted to the data storage layer via Message Queuing Telemetry Transport (MQTT). MQTT 5.0, published by OASIS (New York, NY, USA), was preferred. Signals from the DT platform were stored on Apache Kafka and then forwarded to Apache Flink for analysis. Apache Kafka is an open-source event-streaming platform, whereas Apache Flink is a framework used for stateful computations on unbounded and bounded data streams. Apache Flink version 1.14.1 and Apache Kafka version 3.1, published by Apache Software (Harford County, MD, USA), were determined to be used. In addition, the Unity platform was used for the visualization of the system.

### 3.2. Method and System Design

One of the critical elements in DT is choosing the type of production line. In this study, we chose the automotive industry because its data can be obtained sustainably, it contains many process options and has technological outputs, and can be easily adapted to other areas. The full methodology of the proposed work is shown in Figure 2.

Automotive factories have a large number of new generation tools which are compatible and very beneficial to applying DT. The state-of-the-art tools and equipment are long-lasting and robust so that DT can be performed easily and effectively. The outputs of using DT in the automotive sector can be easily adapted to other fields. For instance, when there are some similar stages in the production of a military land vehicle, this system will be easily shifted to the military field.

For a DT-based system to be successful, it is of great importance that a high amount of data can be obtained in the system to be established, and that the data flow is continuous. In addition, since DT technology is an emerging technology, the fact that the facility where the system is installed has new generation technological elements is another critical factor in gaining the advantages of technology at the maximum level. It is seen that factories with such features in the automotive field in Turkey are predominantly located in and around Bursa. After evaluating all these criteria, one of Turkey’s most important automotive factories operating in Bursa and incorporating emerging technologies in its production processes has been determined.

#### 3.2.1. The Goals in the Automotive Factory

The goals of using DT in the automotive factory regarding the case study are:Increasing productivity by keeping the production line in optimal condition.Making analyses that will increase the life and durability of the produced materials.Enabling an immediate response when an emergency occurs.Testing and managing different scenarios as virtual data can be provided.

The needs of the factory are accelerating production and increasing the profitability and product quality.

#### 3.2.2. Agreed Scenarios

The automotive manufacturing process consists of many sub-processes and parts. During the creation of the digital twin, we preferred scenarios that are used as frequently as possible and where the work outputs can be seen concretely. For this reason, we determined four scenarios at this stage and developed the system based on them. These scenarios are:i.Monitoring and analyzing the temperatures of motors in robotic arms

A study was carried out on the measurement of the temperature of the robotic arms. We performed some analysis on the Parker-MPP series servo motors (Cleveland, OH, USA). Temperature is an important parameter in extending the life cycle of these motors. For example, operating the servo motors at 10 degrees hotter than their maximum temperature can reduce the life of that motor by half. Maximum operating temperatures of servo motors are determined by NEMA (National Electrical Manufacturers Association, Arlington, VA, USA) insulation classes. The insulation class of the motors is chosen as the F class here. These motors must be run above 0 degrees. After determining the boundaries, we identified three regions to carry out our analysis: safe, unsafe, and hazardous region. It can be seen from Figure 3 that the temperature for the safe zones is from 50 to 120. Between 5–50 and 120–150 degrees is unsafe, and below 5 degrees and above 150 degrees, it is a hazardous zone. The classes of motors according to their temperatures are also shown in Figure 3.

ii.Following and analyzing the optimum operating frequency range of the CNC machine

Computer numerical control (CNC) machines, as workbench systems, are systems that perform an automatic operation by programming through the computer mounted on them. Therefore, they are crucial systems. In our system, the frequency and feed values of these machines were very critical in terms of drilling, so they were adjusted to hold the belt to work in optimum conditions. The optimal supply and frequency ranges have been determined as shown in Figure 4.

iii.Monitoring and analyzing the dust density in the factory environment

Dust density is another important factor that affects both human health and the optimum operation of the production line. CNC machines affect the ambient dust density. If the frequency of the CNC machine decreases, the dust density increases with the increase in chip production and damages the machines. For this reason, for the production line to work at the optimum level, the ambient dust density should not exceed 10% (Figure 5). For this, the operating frequency of the CNC machine should not be increased above 1600 rpm for a long time; so, it should be operated in the range of 1500–1550 rpm optimally. To control CNC frequency, an Arduino-based tachometer was used for monitoring and adjusting the frequency.

iv.Monitoring the temperature in the factory environment to detect whether it harms vehicles or sensitive materials

This factory has many robotic arms. Therefore, the first step was determining the optimal ambient temperature conditions for proper processing. Optimal values have been determined as between 0–45 and 10–55 degrees for the Fanuc CRX-10iA robot, and Kuka KR 1000 titan robot, respectively. Note that, for humans, this value is known to be optimum between 18 and 21 degrees. The estimated optimal values of temperature for robot arms are shown in Figure 6. There is no action in this part except for monitoring the temperature of the environment.

#### 3.2.3. System Design

Following the agreement of scenarios, the system design phase was started. The first step in the DT creation process was the data generation layer. In this work, data were received from the live system via sensors.

An E52-ELP6-50-2-A temperature sensor was used for receiving temperature data from the physical system, obtaining the status from the Kuka KSS interface, and the temperature of robotic arms [38]. The Telaire Smart Dust Sensor SM-PWM-1C was used for measuring the dust density, and an Arduino-based tachometer frequency sensor for obtaining the CNC frequency value to hold the frequency in the optimal range [39,40]. The obtained sensor data were transmitted to the data storage layer via Message Queuing Telemetry Transport (MQTT), which is a kind of messaging protocol [41]. There are three sides to the messaging system: sender, receiver, and forwarder. The sending and receiving sides are called “clients”. The forwarding (routing) layer in between is called “MQTT Broker”, which is responsible for delivering the messages from the source to the corresponding targets.

The signals obtained through the sensors were sent to the MQTT Broker, that is, Mosquitto Broker, which was determined through the MQTT protocol. Mosquitto Broker is an open-source message broker application that implements the MQTT protocol. It is frequently used in IoT applications. Incoming MQTT messages are listened to via the signal listening module and then forwarded to PostgreSQL, a central database, to be stored.

Signals from the DT platform were stored on Apache Kafka and then forwarded to Apache Flink for analysis [42,43]. Apache Kafka is an open-source event-streaming platform. It is used by many companies for high-throughput data pipelines, flow analytics, and data analysis. Event streaming can be considered a digital equivalent of the human body’s central nervous system. It is the application of real-time data capture in the form of event streaming from sources such as databases, sensors, mobile devices, cloud services, and software applications. Event streaming thus ensures the continuous flow and analysis of data so that the right information is in the right place and at the right time.

Apache Flink, on the other hand, is a framework used for stateful computations on unbounded and bounded data streams. There are applications such as Hadoop and Spark that are similar to Apache Flink. However, the most important reason for choosing Apache Flink was that it is faster than Hadoop and Spark. In addition, applications can be run at any scale, it can write once and run everywhere, and its in-memory performance stands out compared to its counterparts. Regarding its advantages, Apache Flink was used to making transactions on the decisions and analyses on the DT platform.

As with many new generation applications, one of the most important elements in DT applications is the output, that is, the display of the digital copy. In this work, the Unity platform was used for visualization [44]. One of the most important reasons for using Unity was that it is more suitable for platforms with low hardware specifications such that it enables projects to work on tablets.

The system architecture is shown in Figure 7. Particular attention has been paid to select open-source platforms due to their low-dependent working capability, high audibility and transparency, and availability for co-development and continuous innovation. Furthermore, the errors produced are more likely to be fixed.

Lastly, taking action as a result of the analysis of the system, that is, interfering with the running system, is a controversial issue for production lines. Although the system was designed to trigger automatically after the warnings, our case study left it manually to the human intervention, as preferred by many manufacturers. Hence, the DT-based system in this case study was used as a decision support system. The feedback of this system was producing the warning messages, based on data processing, on the user application interface.

#### 3.2.4. Development of the User Interface Application

In line with the determined system architecture, since correct setup and installation of sensors are some of the most important steps for the success of the DT-based system, the sensors determined in the system design were successfully placed on the live system, and a source was created for data acquisition. After the sensor installations were completed, an interface was designed to display the received data (Figure 8) to control the instantaneous change of data and observe the warning messages.

Sensor types are shown on the left side of the screen, and warning icons on the right side provide information about situations such as product status, loss of profit, and danger. In addition, coloring is used to indicate the optimum state of the system or the level of danger. In addition, as the analysis of the sensor data is being completed, estimations and predictions are displayed on the screen as output messages.

Collecting and analyzing the sensor data received over the system are important steps for the creation of the DT. Since one of the most important outputs of the process of DT is the visualization of the system, by using the unity infrastructure, the production line was visualized (Figure 9).

Visualizations were performed for each element by using color labels from green to red for robotic arm temperature, dust density, CNC frequency, and environment temperature (Figure 10).

With the establishment of the visualization structure, the system installation has been completed. The technical steps in the developed system are given below:The data generated through the sensors’ signal generation module were broadcasted as MQTT messages.Mosquito MQTT Broker grouped the messages received by their headings and forwarded them to web clients.The DT platform received messages with the headers specified. It added information such as date, time, sensor type, and sensor ID and transmitted it to the database communication module for storage.Apache Flink read data from Apache Kafka and performed the analysis. It sent the results of the analysis to the signal listening module to be displayed on the user interface.Unity 3D rea the data stored in PostgreSQL and visualized it in real-time.

## 4. Results and Discussion

The first study was based on measuring the downtime gain by comparing the values before and after using DT. After the system had been successfully created, the determined production line was operated before implementing its DT. Firstly, the determined production line was operated and total downtime data for 6 months period (September 2020–February 2021) were obtained. The results are shown in Table 2.

The system was run by using DT for 6 months (July–December 2021). The results obtained regarding the total downtime are given in Table 3. The comparison of downtime of the system before and after the DT (by months) is shown in Figure 11.

A 992-min gain over 1133 min in downtime, which corresponds to 87.56%, was achieved (Table 4). The comparison of total downtime of the system before and after the DT is shown in Figure 12.

The second study carried out was related to efficiency. Before applying DT technology to the production system, the daily working times of the machines were measured (Table 5).

With the use of DT technology, it was aimed to achieve significant gains in the working time of the machines and increase productivity. As a result, 61,246 min of working time were observed for the same amount of work as a result of running the system obtained after the use of the digital twin in the same period (Table 6). The comparison of the daily working time required to do the same amount of work before and after the DT (by months) is shown in Figure 13.

On the other hand, it has been observed that the efficiency of daily working time was very promising. In total, 65,163 min in the pre-digital twin system has been reduced to 61,246 min with the digital twin-based system, which corresponds to approximately 6.01% gain achieved (Table 7). In this way, both labor and electricity savings were achieved. The comparison of downtime of the system before and after the DT (by months) is shown in Figure 14.

In this process, many difficulties occurred. We encountered some major difficulties during the installation phase of the system. For instance, we first decided to use Apache IoTDB for the collection of sensor data; however, the installation of the infrastructure was problematic. Since Apache IoTDB is also a new platform, there was a lack of sufficient training resources. To overcome this, we applied the “broker + database addition” solution.

Secondly, one of the most challenging issues during operation was the determination and installation of sensor types. Inaccurate choices or installations of the sensors negatively affected the success of the project; a great effort was put forth to identify the type of the sensors and also install them in the right places. While deciding on the sensor types, primarily the production conditions of the factory were taken into consideration. The location conditions of the area where it will be placed on the production line, and the operating conditions of the sensor under conditions such as heat/humidity were evaluated, and the most suitable sensors for these conditions were selected. Sensors with the latest technology, which are thought to be used in future studies, and which have been used in many studies in the past, have been preferred.

Besides obtaining very promising downtime savings and efficiency gains of 87.56% and 6.01%, respectively, we also created a system that enabled the real-time monitoring and control of the system. With the pre-detection of malfunctions, it was possible to prolong the system’s life cycle time and increase the product quality.

## 5. Conclusions

One of the most important key performance indexes (KPI) in production lines is efficiency. Even if a 1% productivity increase is achieved with the optimizations made, it provides significant returns for companies in terms of providing a great income advantage and ensuring competitiveness. In this study, we have shown that DT technology is a revolutionary technology, especially in terms of increasing efficiency, with 6.01%. In addition, an 87.56% increase was achieved in downtime, which is a crucial element of KPIs in production. Although this study has been implemented with a limited number of sensors and predetermined scenarios, the successful results obtained have concretely proven the success of the technology on the production line. It is the first case study in which efficiency increase and false alarm reduction were achieved with the use of DT technology in an extensive automotive production line. With the very promising efficiency and downtime results of this study, it is shown that electricity saving is achieved with high gain which is a good contribution made to the provision of a green society by preventing energy waste. The part of the system which is considered to be improved in future studies is the artificial intelligence analysis part. In this study, the analysis and processing were mostly based on fundamental trend analysis. In future studies, trend analyses are planned to be performed with higher accuracy and diversity by applying artificial intelligence techniques such as machine learning and deep learning. For this purpose, it is evaluated that unsupervised learning and reinforcement learning methods can be used by applying the parameters and input space determined in the first stage.

Since we have many data obtained from this case study, it is suitable to apply an artificial intelligence analysis. As machine learning and deep learning libraries, Tensorflow, Keras, Scipy, and Scikit-learn can be used. In this way, various artificial intelligence-based trend analyses can be made. By using the parameters of the above-mentioned scenarios of machine learning algorithms, it is aimed to determine the target range correctly and to perform the analysis autonomously with higher accuracy. At this stage, rather than using artificial intelligence methods, the activities of establishing the infrastructure and preparing and verifying the training data were carried out.

In addition, we aim to display the digital twin in the AR/VR environment in future studies, where the unity infrastructure would be an appropriate solution.

## Figures and Tables

**Figure 1 sensors-22-06963-f001:**
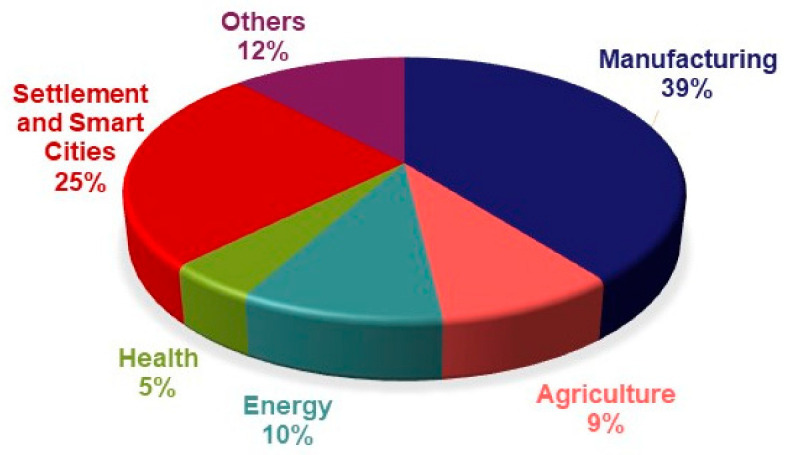
The graph of digital twin market distribution in 2020 [11].

**Figure 2 sensors-22-06963-f002:**
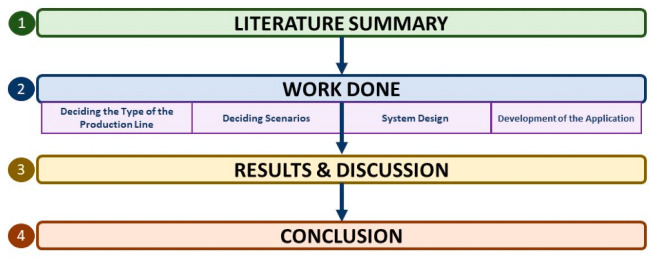
The methodology of the proposed work.

**Figure 3 sensors-22-06963-f003:**
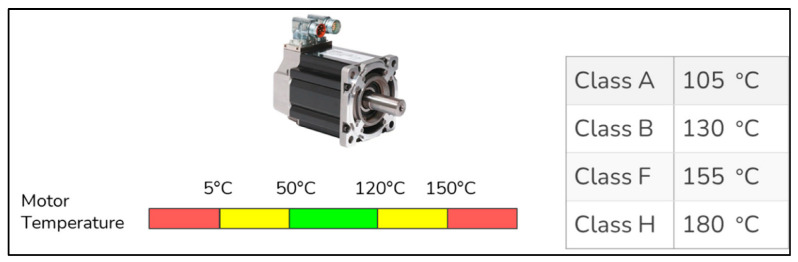
The visual representation of the optimum temperature ranges of the chosen motor.

**Figure 4 sensors-22-06963-f004:**
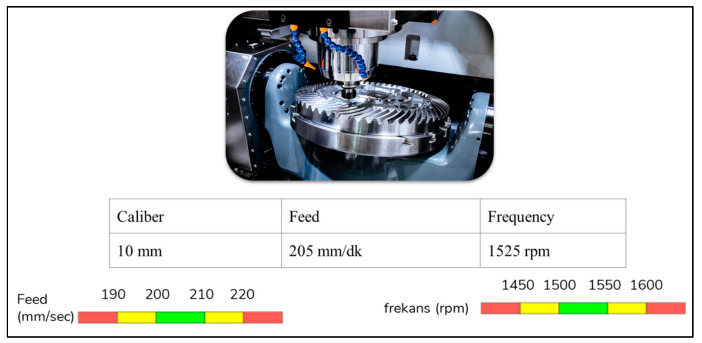
The visual representation of the optimum frequency and feed range of the chosen CNC.

**Figure 5 sensors-22-06963-f005:**
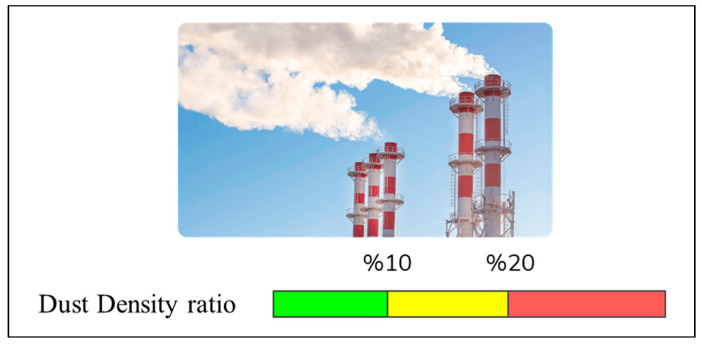
The visual representation of the optimum dust density ratio ranges of the product line environment.

**Figure 6 sensors-22-06963-f006:**
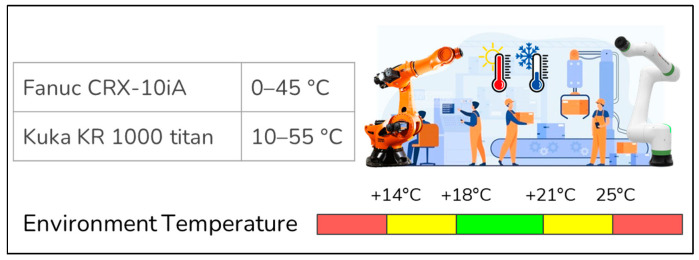
The visual representation of the optimum temperature ranges within the product line environment.

**Figure 7 sensors-22-06963-f007:**
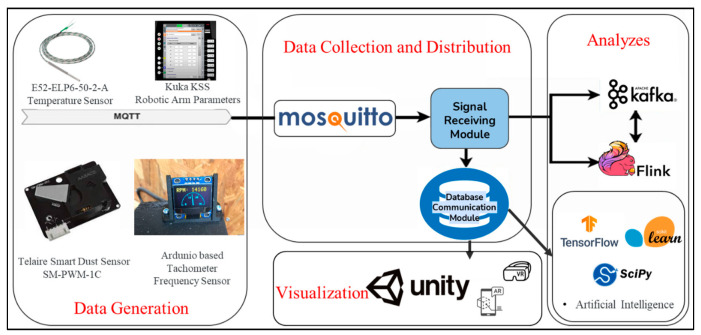
The visual representation of the system architecture.

**Figure 8 sensors-22-06963-f008:**
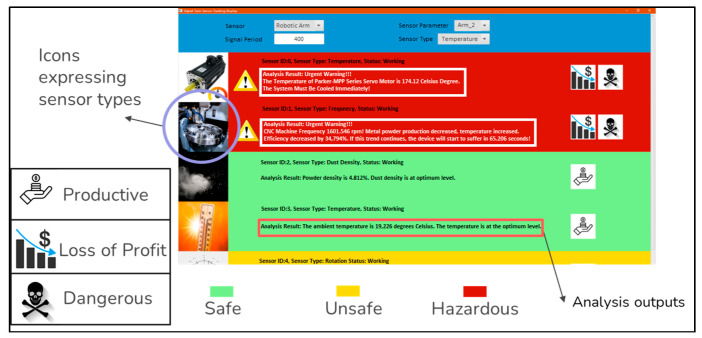
An example of the sensor data and analysis tracking screen.

**Figure 9 sensors-22-06963-f009:**
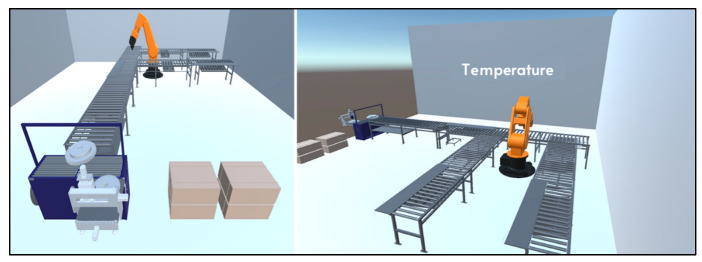
An example of the production line visualization in the system.

**Figure 10 sensors-22-06963-f010:**
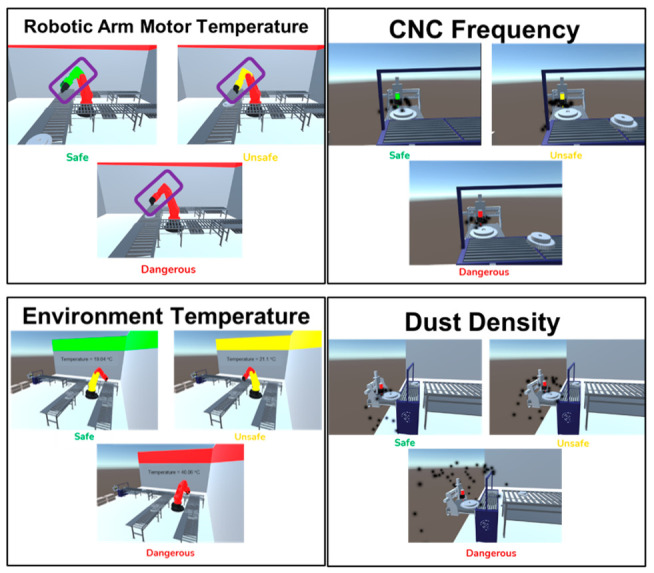
Some screenshot examples of the visualization of all elements in the system.

**Figure 11 sensors-22-06963-f011:**
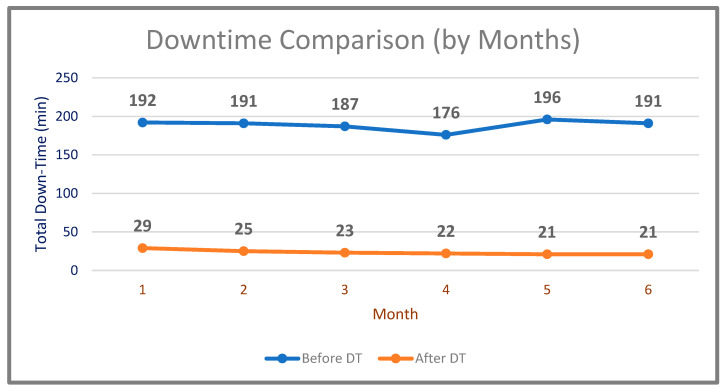
Minutes of downtime comparison by months (before and after DT).

**Figure 12 sensors-22-06963-f012:**
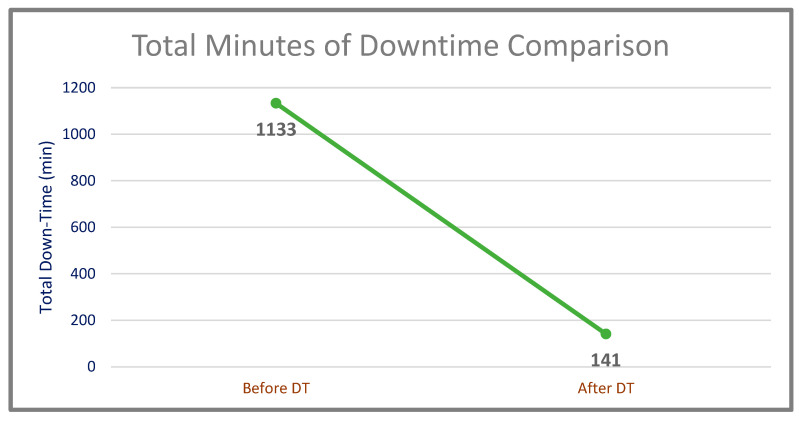
Total minutes of downtime comparison (before and after DT).

**Figure 13 sensors-22-06963-f013:**
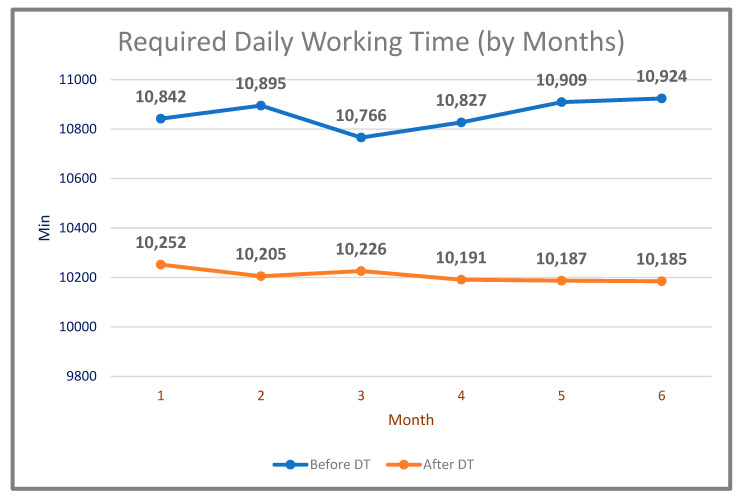
Minutes of daily working time required to do the same amount of work comparison by months (before and after DT).

**Figure 14 sensors-22-06963-f014:**
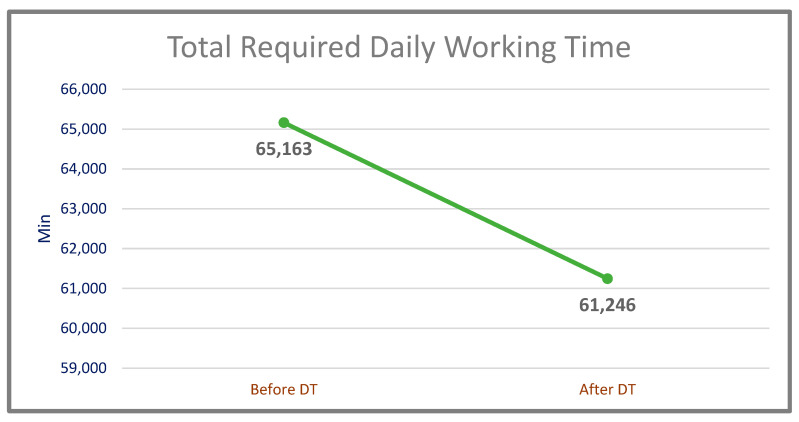
Total daily working time required to do the same amount of work (before and after DT).

**Table 1 sensors-22-06963-t001:** Materials used.

Purpose	Materials
Temperature	E52-ELP6-50-2-A temperature sensor
Dusty Density	Telaire Smart Dust Sensor SM-PWM-1C
CNC Frequency	Arduino-based tachometer frequency sensor

**Table 2 sensors-22-06963-t002:** Downtimes before using DT.

	Total Downtime (min)
September 2020	192
October 2020	191
November 2020	187
December 2020	176
January 2021	196
February 2021	191
Total	1133

**Table 3 sensors-22-06963-t003:** Downtimes after using DT.

	Total Downtime (min)
July 2021	29
August 2021	25
September 2021	23
October 2021	22
November 2021	21
December 2021	21
Total	141

**Table 4 sensors-22-06963-t004:** Downtimes before and after using DT.

	Total Downtime (min)
Before the Digital Twin	1133
After the Digital Twin	141
Total Gain	992

**Table 5 sensors-22-06963-t005:** Daily working time required to do the same amount of work before using the DT.

	Required Time (min)
September 2020	10,842
October 2020	10,895
November 2020	10,766
December 2020	10,827
January 2021	10,909
February 2021	10,924
Total	65,163

**Table 6 sensors-22-06963-t006:** Daily working time required to do the same amount of work after using the DT.

	Required Time to do the Same Amount of Work (Minutes)
July 2021	10,252
August 2021	10,205
September 2021	10,226
October 2021	10,191
November 2021	10,187
December 2021	10,185
Total	61,246

**Table 7 sensors-22-06963-t007:** Daily working time required to do the same amount of work before and after using the digital twin.

	Required Time to do the Same Amount of Work (Minutes)
Before the Digital Twin	65,163
After the Digital Twin	61,246
Total Gain	3917

## Data Availability

The data presented in this study are available on request from the corresponding author.

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
