# Peer review of "A Digital Twin Case Study on Automotive Production Line"

_sensors, 2022, doi:10.3390/s22186963_

Round 1
Reviewer 1 Report (Previous Reviewer 2)
The authors have addressed all the comments suggested to them. But the graphs which are added newly can be improved if possible. Make them attractive, including data labels, so they may be easy to understand.
Otherwise, the paper can be accepted in the present form.
Author Response
Please see the attachment.

Reviewer 2 Report (Previous Reviewer 3)
Language of the manuscript-at-hand still needs to be revisited. For instance, Lines 150 - 152 states, In Cyber-Physical Systems where controllers are often hard (where controllers are often hard?) real-time data collection for digital twins is a challenge ...
Quality of the Figure 7 needs to be enhanced. The same is true for Figure 8 and Figure 9.
Contributions have been incorporated in Section 1, Introduction, already (Lines 97 - 103). There is perhaps no need to rewrite them at the end of the Section 2, Literature Review. In fact, the ones delineated in Section 1, Introduction, could be written in sequential manner as (1) ... , (2) ... , (3) ... .
Author Response
Please see the attachment.

This manuscript is a resubmission of an earlier submission. The following is a list of the peer review reports and author responses from that submission.
Round 1
Reviewer 1 Report
This study presents the use of digital twin technology in the commercial production phase of the automotive production line with a case study. The evaluation results are presented.
The paper presents an interesting research topic with an industrial application. It is well-structured and well-written. However, the paper needs improvement. Some of my suggestions are as follows:
First, the paper needs thorough proofreading and language improvement.
Also, the literature study needs to be improved. The recent studies need to be considered. Also, some academic studies in this field relate to a production line.
As another aspect of the literature review, I believe you need to address DT for CPS and IoT fields enabling technologies. Example study related to this is as follows:
@inproceedings{mertens2020towards,
title={Towards real-time cyber-physical systems instrumentation for creating digital twins},
author={Mertens, Joost and Challenger, Moharram and Vanherpen, Ken and Denil, Joachim},
booktitle={2020 Spring Simulation Conference (SpringSim)},
pages={1--12},
year={2020},
organization={IEEE}
}
Finally, industrial application in your study is a plus for your work. However, the scenario in the automotive production line needs to be elaborated on in more detail.
Reviewer 2 Report
The authors have studied the role of the digital twin in the automotive production line with the case study. Although the authors have broadly covered all the aspects and explained them clearly, the paper needs some modifications before possible publication.
1. Line 9: Electrics, besides, it should be electronics.
2. Line 22-24: Repeated use of 'with.' Kindly restructure both sentences to maintain the interest of the reader.
3. Line 37: "DT is an electronic copy of a physical entity." DT is not just an electronic copy; it also includes digital (or virtual representation) of the physical entity. So, it is not an appropriate choice to use the word electronic copy.
4. Besides, mentioning materials, and methods a therotically, explain them in tabular form.
5. Are the figures copied or redrawn from the source? If so, kindly take the necessary permission from the author of that study.
6. Section 4: The various comparisons are made for the effect of DT on downtimes. Use the graphs for these comparisons, which will ease the analysis.
7. Some aspects of DT are missing in the introduction regarding the Communication protocol, machine learning, and architecture of DT. Kindly refer to the following recent articles to modify the introduction part (Reference not limited to this you can refer more if possible)
https://doi.org/10.3390/su131810139
10.1109/ACCESS.2017.2657006
https://doi.org/10.1016/j.aei.2020.101225
8. The paper also requires a minor English grammar and spell-check.
9. Highlight the uniqness in the study carried out with respect to the previous study done.
Reviewer 3 Report
The contributions of the manuscript-at-hand should have been delineated in a sequential manner. In fact, authors have delineated the same towards the end of Section 2 (Literature Review), however, the same should be in fact moved at end of Section 1 (Introduction).
Language and Sentence Structure of the manuscript-at-hand needs significant improvement (... With the extension to manufacturing, prognostics, design, and health management [9][10][11], It ('It' should be changed to 'it') is ...).
Also, a comparison of the studies delineated in Section 2 (Literature Review) is indispensable and should be presented in the form of a Table that outlines the salient charcterisics, pros, and their cons.
Looking at the Section 3 (Materials, Method, and Applications) and Section 4 (Results and Discussion), it has become aparent that the manuscript-at-hand has perhaps originated from a Project Report and, therefore, lacks depth that is indispensable for a research paper. In fact, the Results presented in Section 4 are mere in the aspect of the 'Time (In Minutes)' and nothing more.